# Remote Management of Heart Failure in Patients with Implantable Devices

**DOI:** 10.3390/diagnostics14222554

**Published:** 2024-11-14

**Authors:** Luca Santini, Francesco Adamo, Karim Mahfouz, Carlo Colaiaco, Ilaria Finamora, Carmine De Lucia, Nicola Danisi, Stefania Gentile, Claudia Sorrentino, Maria Grazia Romano, Luca Sangiovanni, Alessio Nardini, Fabrizio Ammirati

**Affiliations:** 1Cardiology Department, GB Grassi Hospital, ASL Roma 3, 00122 Rome, Italy; lucasantinimd@gmail.com (L.S.); karim.mahfouz@aslroma3.it (K.M.); carlo.colaiaco@aslroma3.it (C.C.); ilaria.finamora@aslroma3.it (I.F.); carmine.delucia@aslroma3.it (C.D.L.); nicola.danisi@aslroma3.it (N.D.); stefania.gentile@aslroma3.it (S.G.); claudia.sorrentino@aslroma3.it (C.S.); mariagrazia.romano@aslroma3.it (M.G.R.); luca.sangiovanni@aslroma3.it (L.S.); fabrizio.ammirati@aslroma3.it (F.A.); 2General Direction of the Mission Unit for the Implementation of PNRR Interventions, Italian Ministry of Health, 00144 Roma, Italy; a.nardini@sanita.it

**Keywords:** heart failure, remote monitoring, CIEDs, HeartLogic, HeartInsight, Triage-HF, CardioMEMS, telemedicine, digital medicine

## Abstract

**Background**: Heart failure (HF) is a chronic disease with a steadily increasing prevalence, high mortality, and social and economic costs. Furthermore, every hospitalization for acute HF is associated with worsening prognosis and reduced life expectancy. In order to prevent hospitalizations, it would be useful to have instruments that can predict them well in advance. **Methods**: We performed a review on remote monitoring of heart failure through implantable devices. **Results**: Precise multi-parameter algorithms, available for ICD and CRT-D patients, have been created, which also use artificial intelligence and are able to predict a new heart failure event more than 30 days in advance. There are also implantable pulmonary artery devices that can predict hospitalizations and reduce the impact of heart failure. The proper organization of transmission and alert management is crucial for clinical success in using these tools. **Conclusions**: The full implementation of remote monitoring of implantable devices, and in particular, the use of new algorithms for the prediction of acute heart failure episodes, represents a huge challenge but also a huge opportunity.

## 1. Introduction

Heart failure (HF) is a chronic disease with a steadily increasing prevalence, high mortality, and social and economic costs. Furthermore, every hospitalization for acute HF is associated with worsening prognosis and reduced life expectancy [1] (Figure 1). Patients require frequent clinical monitoring with high commitment on the part of healthcare facilities to avoid flare-ups and new hospitalizations. Many patients affected by HF wear cardiac implantable devices (CIEDs).

Over the last 20 years, remote monitoring (RM) of implantable devices has progressively evolved from simple device monitoring to patient monitoring (Figure 2).

Initially, HF monitoring was performed by transthoracic impedance measurement; however, this strategy led to a large number of false positives and did not lead to a reduction in mortality and re-hospitalizations [2].

A big step forward has been made thanks to digital medicine and especially to the implementation of artificial intelligence (AI) algorithms on implantable devices. The new generations of CIEDs, indeed, are capable of providing a multi-parametric evaluation that goes beyond the single monitoring of transthoracic impedance, integrating it with several other parameters.

Furthermore, many ICDs are nowadays equipped with automated algorithms able to generate a composite risk score or to define a risk profile, which allows us to predict episodes of HF flare-ups well in advance with the aim to reduce hospitalizations and their related costs [3,4].

We therefore conducted a review of remote monitoring systems for heart failure, including clinical trials and real-world studies.

## 2. New Multi-Parametric Algorithms

HF remote monitoring can be performed by pulmonary artery standalone implantable devices capable of detecting pulmonary arterial pressure (e.g., CardioMEMS) or by multi-parameter algorithms based on artificial intelligence integrated in the newest models of CIEDs (ICDs, CRTs, some pacemakers). The latter has the advantage of avoiding the implantation of additional devices in patients who already often wear CIEDs due to their underlying pathology.

These algorithms can detect the early warning signs of worsening heart failure from a wide set of sensors chosen to assess the different aspects of HF pathophysiology (Figure 3). The monitored parameters include cardiac tones (monitored by the mean of an accelerometer that is in the CIED pulse generator), heart rate, HRV (heart rate variability), intrathoracic impedance, respiratory rate and volume, physical activity, ventricular extrasystole burden, atrial fibrillation burden, mean HR during atrial fibrillation (AF), and biventricular pacing rate (Table 1).

These collected data are integrated by mathematical models to incorporate multiple sensors in a single composite index. When the index overcomes the established threshold, an alert is generated to advise, through the remote monitoring platform, that an acute event is likely to occur in the following 30/40 days or to identify decompensation risk ranges on an individual basis.

The preliminary experiences in clinical practice show that these automated algorithms are very promising and extremely well performing, although there are not yet strong data on reduced mortality and hospitalizations. However, the interest of the scientific community and manufacturers in this field is very strong, and in the coming years there will likely be new evidence and further technological developments able to improve these algorithms and hopefully achieve stronger end points.

Most CIED manufacturers have developed their own algorithms. Each includes specific parameters for multi-parametric evaluation and provides for the use of a composite index or risk intervals.

### 2.1. Heart Logic (Boston Scientific)

The HeartLogic™ algorithm (Boston Scientific, St. Paul, MN, USA) is available via the Latitude remote monitoring platform in patients with the newest generations of ICD and CRT-D devices. It uses five sensors to detect the incipient episode of acute heart failure: first heart sound (S1, associated with ventricular contraction status and reduced by contractility reduction) and third heart sound (S3, associated with early diastolic filling), S3/S1 ratio (increased during acute heart failure because of S1 reduction and S3 increase), respiratory rate and volume (rapid shallow breathing patterns associated with shortness of breath), intrathoracic impedance (associated with fluid accumulation and pulmonary edema), nocturnal heart rate (an indicator of cardiac status), and amount of physical activity (global patient status and fatigue).

The integration of these parameters leads to the HeartLogic™ index, whose nominal threshold value is 16. An increase above the threshold represents an alert for possible decompensation.

The largest study in which HeartLogic™ was investigated is the Multisense study [5].

The MultiSENSE was an international, multicenter, non-randomized feasibility study that enrolled more than 900 patients with CRTD and validated that the HeartLogic algorithm provides a sensitive and timely predictor of impending heart failure decompensation.

The sensitivity for HF recurrences (HF admissions or unscheduled visits requiring intravenous treatment) of an alert triggered by a HeartLogic™ index value of 16 or higher was 70% (95% confidence interval, CI: 55.4% to 82.1%) with an unexplained alert rate of 1.47 per patient-year (95% CI: 1.32 to 1.65). The median time between an alert and the following HF event was 34.0 days (interquartile range: 19.0 to 66.3 days). Every sensor showed a different average change during heart failure events (Figure 4).

A post hoc analysis of the MULTISENSE trial by Gardner et al. provided further data on the usefulness of the index [6]. Patients with a high baseline index had a 10-fold higher risk of decompensation as compared with patients with an index value <16. Moreover, Gardner et al. demonstrated that the prognostic value of the index was independent from the value of the n-terminal pro B-type natriuretic peptide (NT-proBNP); when compared with patients with low NT-proBNP (<1000 pg/mL) and a negative HeartLogic™ index, patients ‘in alert’ had a 24-fold and a 50-fold increased risk of HF events at follow-up if NT-proBNP levels were low (<1000 pg/mL) or increased (>1000 pg/mL), respectively [6].

A subsequent prospective study showed a low rate of unexplained alerts and the presence of clinically detectable signs of HF in only 34% of patients with active alerts. [7]

Another study reported that the burden of HeartLogic™ alerts was similar between CRT-D and ICD patients, while patients with AF and CKD seemed more exposed to alerts, but the ability of the HeartLogic algorithm to identify periods of significantly increased risk of clinical events was confirmed, regardless of the type of device and the presence of AF or CKD [8].

MANAGE-HF phase 1 enrolled 200 patients implanted with a CRT-D or ICD enabled with HeartLogic™ [9]. The study found that HeartLogic was safely integrated into clinical practice and associated with lower natriuretic peptide levels and hospitalization rates.

MANAGE-HF phase 2 was cancelled in agreement with the FDA because the results would arrive too late and the enrolling centers would not randomize to no-HeartLogic [10]. The study was replaced by DANLOGIC-HF [11]. The purpose of the study was to assess the impact of HeartLogic-guided management on clinical outcomes among patients implanted with a cardiac device.

The study is a pragmatic, registry-based, randomized controlled trial. Eligible patients in Denmark will be randomized 1:1 to either HeartLogic-guided management or usual care. HeartLogic alerts will be managed according to a prespecified management guide. Estimated study completion is December 2027.

Further studies showed that HeartLogic is sensible to the onset of atrial fibrillation [12], may detect sub-optimal BiV pacing and poor physiologic response [13], and may identify patients vulnerable to RV pacing [14].

Finally, more studies and real-world data have demonstrated the usefulness of HeartLogic, also in terms of the prediction of arrhythmia and reduction in economic costs and resources [15,16,17,18,19,20,21,22].

### 2.2. HeartInsight (Biotronik)

HeartInsight (BIOTRONIK SE&Co. KG, Berlin, Germany) is a recently introduced algorithm available in the remote monitoring platform (Home Monitoring) of ICDs and CRT-Ds manufactured by Biotronik.

HeartInsight uses the detection of seven parameters (mean heart rate, mean heart rate at rest, premature ventricular contractions, atrial burden, heart rate variability, patient activity, thoracic impedance). The longitudinal variations in these parameters are daily recorded and combined with each other and (optionally) with a baseline risk stratifier, the Seattle HF model (based on demographic, clinical, therapy, blood, and urine data), in order to improve specificity [23].

The combination of these parameters, after a rolling window, generates the HeartInsight index, the alarm threshold of which is programmable (default setting: 45) according to the patient’s characteristics, and when the threshold is exceeded, an alert is generated.

The main study on this algorithm is the Selene HF that enrolled 918 ICD or CRT-D patients, randomly allocated in the algorithm derivation group or in the algorithm validation group [24].

The receiver operating characteristic curve for HF hospitalization was 0.89, and in the latter sensitivity of predicting HF hospitalizations was 65.5%.

The median time between an alert and the following HF event was 42 days (interquartile range 21–89), and the false (or unexplained) alert rate was 0.69 (CI 0.64–0.74) per patient-year.

When the Seattle HF model was removed from the algorithm, sensitivity did not change, while the false alert rate increased to 0.76/patient-year.

There was a low alert burden, with <1 alert per patient-year.

A meta-analysis including Selene HF and other previous studies in which there were patients with active Home Monitoring Biotronik evaluated 2050 patients, showing that the mean HeartInsight score was significantly higher at 12 weeks before worsening heart failure hospitalization (WHFH) than in the no WHFH group, and it further increased by 22% until a WHFH event [25].

Further real-world data confirmed the usefulness of the HeartInsight algorithm [26].

#### TRIAGE HF (Medtronic)

Triage HF (Medtronic Inc., Minneapolis, MN, USA) is an algorithm available in all CIEDs manufactured by Medtronic and equipped with Optivol™ (an algorithm based on thoracic impedance).

It integrates multiple parameters (Optivol, patient activity, burden of atrial fibrillation, ventricular rate during atrial fibrillation, ventricular pacing rate, nocturnal heart rate, HRV, shocks delivered, treated episodes of VT/VF).

The main difference with the already mentioned algorithms is that they do not compute a numerical index but a score of risk of hospitalization for acute HF in the next 30 days divided into HIGH, MEDIUM, and LOW. Based on a Baynesian probabilistic model, they use 30 days of diagnostic data to predict the risk of decompensation in the next 30 days. The risk score is ranked according to the highest risk in the last 30 days. It is calculated upon receipt of data on Carelink (Medtronic Inc., Minneapolis, MN, USA). The first score is available at 65 days after implantation (34 days for Optivol calibration and 30 days for TriageHF risk stratification).

The first study to evaluate the algorithm was PARTNERS-HF (Program to Access and Review Trending Information and Evaluate Correlation to Symptoms in Patients with Heart Failure), which enrolled 694 CRT-D patients [27]. Patients with positive combined HF device diagnostics had a 5.5-fold increased risk of hospitalization for acute HF in the next 30 days. The risk remained high (hazard ratio 4.8; 95% confidence interval: 2.9 to 8.1, *p* < 0.0001) even after adjusting for clinical variables.

Next, studies showed that the algorithm, by evaluating changes in these device diagnostic parameters, improved the ability to identify patients at risk of HF events in a 30-day period, with the possibility of classifying patients as exposed to a high, medium, or low risk of HF events [28,29].

The TRIAGE-HF study enrolled 100 patients with systolic HF and wearers of Medtronic CIEDs, with an 8-month follow-up [30].

Twenty-four high heart failure risk status (HFRS) occurrences were observed among 100 subjects.

Device parameters associated with increased risk of HF hospitalization included OptiVol index (*n* = 20), followed by low patient activity (*n* = 18) and elevated night heart rate (*n* = 12).

The finding of high HFRS was associated with the occurrence of worsening heart failure symptoms in 63% of the cases, which increased to 83% when non-compliance with drug therapy and lifestyle was considered.

Actions were taken in 54% of high-risk alerts. High device-generated HF risk status showed good sensitivity (98.6%) and average specificity (63.4%) in identifying worsening HF events [31].

TRIAGE HF data can be integrated with patient-reported outcomes (reported by the patient in the MYTRIAGEHF Patient App) in the Angels of HF report available on Carelink.

## 3. Pulmonary Pressure Monitoring Devices

Another strategy for remote monitoring of heart failure is the measurement of pulmonary pressure. Monitoring of pulmonary artery pressure using a wireless hemodynamic monitoring system is also suggested with indication IIb level B recommendation by the 2021 ESC guidelines for the diagnosis and treatment of acute and chronic heart failure [32].

The only device currently approved is the CardioMEMS™ system (Abbott, Abbott Park, IL, USA).

It consists of an implantable sensor in the pulmonary artery that, through a patient electronic unit (a special pillow), communicates with the Merlin.net remote monitoring platform, transmitting pulmonary arterial pressure values (Figure 5).

The sensor measures PA pressures using microelectromechanical system technology with a piezoelectrical membrane (wireless). Distortion of the piezoelectrical membrane in the sensor (in the vessel) changes the resonance frequency signal, which corresponds to a pressure shift. These changes are a surrogate measure for fluid retention in the lungs caused by worsening heart failure. The sensor is implanted in a branch of the left pulmonary artery through the femoral vein. The sensor is powered by radio frequency (RF) energy and does not require replacement due to battery depletion.

The first trial to study CardioMEMS™ was the CardioMEMS Heart Sensor Allows Monitoring of Pressure to Improve Outcomes in NYHA Class III Heart Failure Patients (CHAMPION) Trial, published in 2011, which studied 550 patients with chronic HF in NYHA functional class III with one previous HF hospitalization in the past 12 months from 64 participating centers in the USA [33]. Patients were randomized to the CardioMEMS HF system (*n* = 270) compared to the control group (*n* = 280) and were studied for at least 6 months. It showed significant reduction in HF hospitalizations of 28% (HR 0.72, 95% CI 0.60–0.85; *p* < 0.0001), with an excellent safety profile.

The MEMS-HF study was a prospective, non-randomized, multicenter post-marketing study conducted at sites in Germany, the Netherlands, and Ireland, published in 2020 [34]. It showed a 62% reduction in mortality at one year using CardioMEMS, also confirming the safety of the device. It has also shown an improvement in the quality of life of patients, analyzed through the Kansas City Cardiomyopathy Questionnaire (KCCQ).

The Post-Approval Study (PAS) was a multicenter prospective open-label study performed in 1200 chronic HF patients with a prior HF admission within 12 months from 104 sites from the United States and published in 2020 [35]. Patients were selected irrespective of their HF subtype and ejection fraction percentage.

The rate of HF hospitalization was significantly reduced with PA monitoring by the CardioMEMS HF system by 57% (HR 0.43; 95% CI 0.39–0.47, *p* < 0.0001).

A similar non-randomized, open-label study, the CardioMEMS HF System Post-Market Study (COAST), showed preliminary results in the first 100 patients superimposed on those of the other studies [36].

In 2021, the results of the randomized haemodynamic-GUIDEed management of heart failure (GUIDE-HF) study were published [37].

In this multicenter study, 1000 patients with both chronic HFrEF and HFpEF and successfully implanted with a pulmonary artery pressure monitor were randomized to hemodynamic-guided HF management or a usual care control group. After 12 months of follow-up, it has been shown that hemodynamic-guided management was not superior to the control group in terms of a lower composite endpoint rate of mortality and total HF events. However, follow-up was affected by the COVID-19 pandemic, and a pre-COVID-19 impact analysis showed a lower HF hospitalization rate compared with the control group, suggesting a possible benefit of hemodynamic-guided management.

A later study to GUIDE-HF is MONITOR HF (Hemodynamic Monitoring In Heart Failure), an open-label randomized trial conducted in 25 centers in the Netherlands [24,38]. Eligible patients had New York Heart Association class III chronic heart failure and a previous hospitalization for heart failure, regardless of ejection fraction. Patients were randomly assigned (1:1) to CARDIOMEMS or standard care.

It showed a significant improvement in quality of life, analyzed through the KCCQ, and a reduction in hospitalizations.

Another implantable pulmonary pressure monitoring system is the Cordella™ PA sensor system, which recently received FDA premarket approval.

## 4. Discussion

Despite significant advances in medical and device therapy over the past 30 years, heart failure remains one of the greatest clinical problems of our time, featured by a very high morbidity, mortality, and associated economic burden. Data from the literature show that the risk of death increases with each hospitalization for acute decompensation. In the effect study, the risk of death increased from 2.4 times after the first admission for decompensation to 5.2 times after the fourth admission [39].

Thus, the number of repeat admissions for HF is a strong predictor of mortality among patients with HF.

One of the main objectives in the management of patients with HF should therefore be to reduce hospitalizations for flare-ups. The only way to achieve such a target, in addition to providing optimized pharmacological and non-pharmacological therapy, is to follow patients with close follow-ups to be able to make an early diagnosis and intervene appropriately and possibly before the symptoms of relapse appear.

Considering the ever-increasing prevalence of heart failure due to the aging of the population and the concomitant constant reduction in economic and human resources available to support healthcare systems, the target to achieve a strict and personalized follow-up of patients with HF appears to be too ambitious and well out of reach. This dangerous mismatch can only be addressed through the implementation of digital healthcare in clinical practice.

Digital health, indeed, responds exactly to what new generations of patients demand from technology. Their needs include being followed up remotely as frequently as possible by a safe and effective therapeutic strategy and achieving autonomy and freedom from hospital access and hospitalization. Technological support should warrant them to live better with their chronic diseases.

Remote control of implantable devices since the early 2000s has been the true precursor of modern telecardiology.

Since its first implementation in clinical practice, it was overt that we could shift from simple remote device monitoring to complete remote patient management, as most detected events are clinical events.

Modern devices have a wide variety of sensors on board that are always active and capable of registering changes in various clinical parameters related to worsening compensation status at an early stage, thus enabling the physician to take preventive action.

Nowadays, these devices are therefore able to provide information comparable to most outpatient clinical evaluations and allow for a multi-parametric evaluation, which we know is crucial for obtaining a reliable prediction. In fact, all studies based on transthoracic impedance monitoring alone as the only clinical parameter have had neutral or disappointing results, documenting low sensitivity (ranging between 20 and 30%), a low positive predictive value, and a high annual false-positive rate [40,41,42].

The first study to show an important benefit of RM in terms of reduction in all-cause mortality was IN-TIME [43]. The IN-TIME study demonstrated, indeed, improved clinical status for HF by implementing a RM system based on a reliable transmission rate of 85% and a clinically relevant set of clinical and technical parameters. However, these results required the presence of a hub monitoring center and a clinical workflow that enables fast patient contact and follow-up within one to two working days through a dedicated staff.

More recently, several studies evaluating the use of new algorithms capable of providing a multi-parameter assessment have shown how, with the help of artificial intelligence, it is possible to predict a decompensation event [5,24,27].

Analyzing the data reported in the literature, the most important results we can glean from the adoption of these algorithms in clinical practice are the following:

Early warning time for hospitalizations (median alerting time up to 42 days).

The majority of alerts occurred after the discontinuation of or reduction in prescribed HF therapy. Therefore, they proved to be extremely useful to improve adherence to HF therapy.

The alert-based remote management strategy is more efficient than scheduled follow up. This issue is a key factor in optimizing the available resources and better organizing HF units and remote monitoring clinics.

The general rate of alerts is quite low; thus, adoption of an alert-based management strategy would not generate a high workload at the centers, resulting in a more efficient use of healthcare resources.

HF event rate is significantly higher in alert vs. out of alert. This means that we can stratify our patients, giving priority to those in an alert status. Once more, this results in a fundamental optimization of our limited resources.

Taking clinical actions in response to the alert is associated with a lower risk of HF events. Post-alert interventions allow early awareness of worsening heart failure (proactive approach), increasing compliance with therapy, reducing hospitalizations, and ultimately improving outcomes.

However, changes are needed in the organization of centers and in the way remote monitoring is carried out to maximize the usefulness of these systems. We still face, indeed, some open issues and challenges as follows:

How to react to an alert? There is still a lack of standardization of alert reaction, both in terms of timing and actions to be undertaken. Every alert must be carefully analyzed, and every patient with an alarm must be carefully assessed (device programming, compliance with prescribed therapy, physical activity, etc.). Moreover, there is the need for a learning curve in the management of the alert, especially in asymptomatic patients. Enhanced decompensation therapy should be considered even in the absence of symptoms [4]. The goal should be to switch from reactive medicine to proactive medicine.

An alert-based strategy should be implemented as suggested by guidelines, but in order to accomplish this aim, solutions aimed at improving patient enrollment and ensuring the maintenance of regular connectivity are imperative [44].

Substantial reduction in the burden of nonactionable remote and in-office visits should be pursued. Carrying out some simple changes in the number of scheduled transmissions and optimizing the programming of the devices makes it possible to reduce the number of transmissions and makes the remote monitoring of the devices more sustainable [45].

Customized alert programming based on clinical indications and avoidance of nonactionable alerts are key factors as well.

Finally, according to guidelines, manufacturers may play a fundamental role in supporting optimization of CIEDs’ remote monitoring [44]. They can indeed contribute to facilitating enrolment of the RM platform to ensure regular connectivity, providing adequate training to properly program remote alerts specific to the clinical indication.

Structured organizational models of RM clinics are mandatory. Dedicated staff, including technicians, nurses, physicians, and ancillary staff, should be available, and clear roles and duties should be addressed (who does what). Moreover, the optimal implementation of RM for the management of HF patients requires a multidisciplinary team active along the entire HF pathway. An optimal workflow should include a primary nursing model and integration between HF clinicians and electrophysiologists. According to the specific capabilities and available resources, the model that best suits the center should be implemented.

Patient engagement is another fundamental step. Patient education, in its essential elements of empowerment and engagement, is a continuous and essential process to improve patient compliance and the effectiveness of remote monitoring. The key element is proper communication, whose correct modalities should be formalized and standardized as soon as possible. Communicating with the patient is crucial, providing little information but centered on the problem. A dedicated app may be very useful to establish bidirectional communication between centers and patients and caregivers. For example, a notification through the app can remind the patient to regularly take drug therapy or to ask the patient simple questions to assess the state of compensation.

The current RM platforms of the various companies do not allow full integration with the data of the clinical records. When we receive an alert, we do not always have all the clinical data we need to quickly manage it. We would need a single integrated hospital platform that would allow us to always have at our fingertips the personal, clinical, procedural, and instrumental data, the current therapy, the archive of device checks, and previous actions taken.

In many countries we still face with legal and administrative issues and lack of reimbursement for RM and consequently of resources. Privacy, data integrity and protection, professional responsibility, and liability are other important factors that need to be addressed.

## 5. Conclusions

The full implementation of remote monitoring of implantable devices and, in particular, the use of new algorithms for the prediction of acute heart failure episodes represents a huge challenge but also a huge opportunity. In the future, RM will surely integrate data collected by implanted devices with data collected by external sensors. Fully automated integration of data on device status, blood tests, and hemodynamic and clinical parameters can also be expected within a few years.

Telecardiology (with devices but also non-invasive), if combined with a strict and efficient organizational model and with the increasingly real integration of AI, can significantly improve the management of HF.

Current evidence shows that, thanks to recent technological developments, we can predict heart failure flare-ups. To know whether we will also be able to PREVENT exacerbations and reduce hospitalizations, we still have to wait for the results of randomized trials and to gain more experience in real life.

## Figures and Tables

**Figure 1 diagnostics-14-02554-f001:**
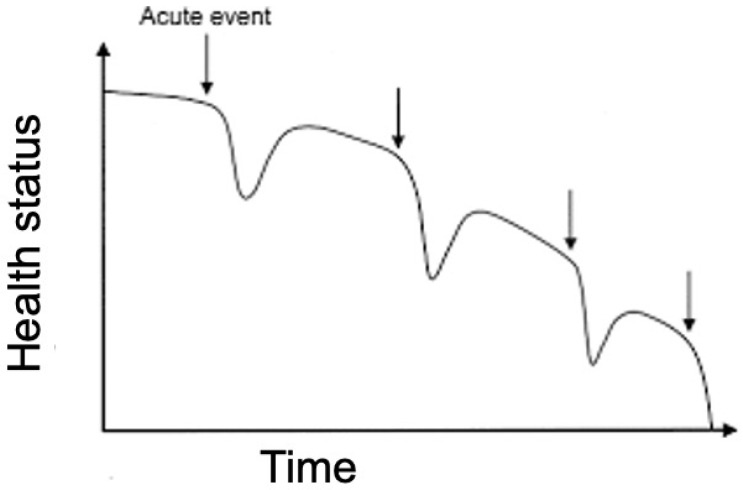
Progression of worsening heart failure [1].

**Figure 2 diagnostics-14-02554-f002:**
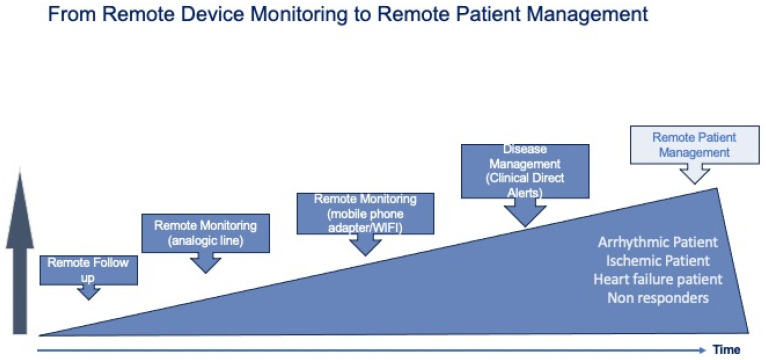
From device monitoring to remote patient management.

**Figure 3 diagnostics-14-02554-f003:**
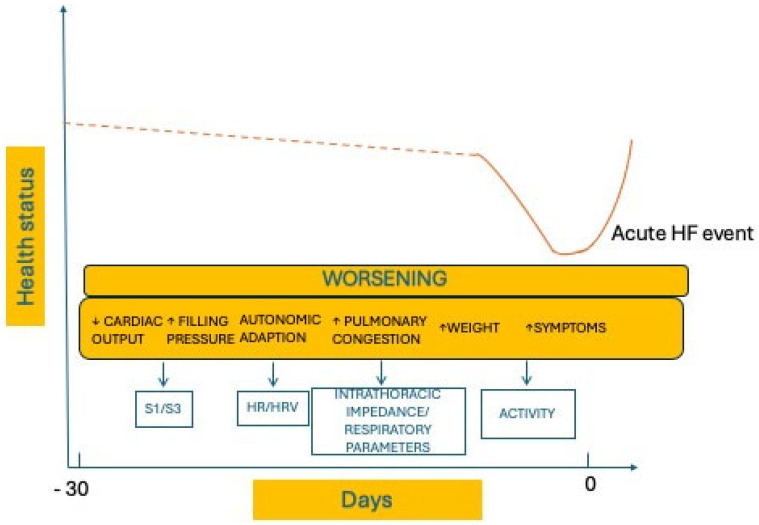
Pathophysiological course of an episode of acute heart failure in the month preceding the symptoms, resulting in changes in heart tones, heart rate, heart rate variability, intrathoracic impedance, respiratory parameters, and physical activity. These changes can be detected by new algorithms that can predict an episode of acute heart failure well in advance. S1, first heart sound; S3, third heart sound; HR, heart rate; HRV, heart rate variability [4].

**Figure 4 diagnostics-14-02554-f004:**
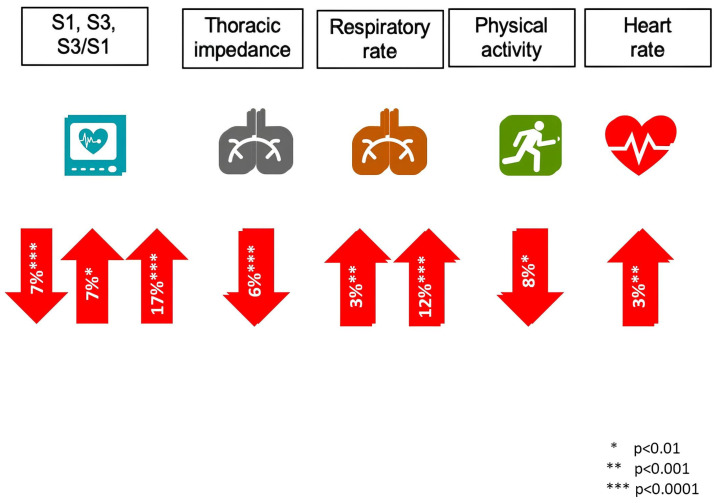
Average sensor value changes in patients with a heart failure event (matched paired analysis) from the Multisense study.

**Figure 5 diagnostics-14-02554-f005:**
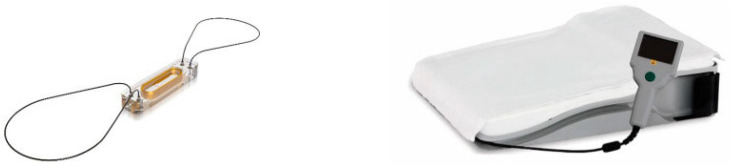
The CardioMEMS pulmonary artery pressure sensor and patient electronics unit. Figures used with permission from Abbott.

**Table 1 diagnostics-14-02554-t001:** Parameters analyzed by each algorithm for the prediction of acute heart failure episodes. PVC, premature ventricular contractions; VT/VF, ventricular tachycardia, ventricular fibrillation.

HeartLogic	HeartInsight	Triage-HF
S1, S3, S3/S1	Mean heart rate	Optivol (intrathoracic impedance)
Respiratory rate	Mean heart rate at rest	Burden of atrial fibrillation
Intrathoracic impedance	PVC burden	Ventricular rate during atrial fibrillation
Nocturnal heart rate	Atrial burden	Ventricular pacing rate
Amount of physical activity	Heart rate variability	Nocturnal heart rate
	Patient activity	Heart rate variability
	Intrathoracic impedance	Shocks delivered
	Seattle HF model (optional)	Treated episodes of VT/VF

## Data Availability

Not applicable.

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
