# Peer review of "Remote Management of Heart Failure in Patients with Implantable Devices"

_diagnostics, 2024, doi:10.3390/diagnostics14222554_

Round 1
Reviewer 1 Report
Comments and Suggestions for Authors
I congratulate the Authors for the interesting as comprehensive review that deals with a growing Medical as social problem.
After a discussion of the state of the art of the technology available to date to remotely monitor patients suffering from heart failure, the Authors move on to substantive considerations regarding the change in organisational care logic. This part is undoubtedly the most interesting and thought-provoking for readers.
A small personal note: I would reduce the description of the individual systems as much as possible, also to avoid mixing commercial aspects with purely technological ones, also because of the rapid obsolescence of biomedical technology.
Author Response
Thank you for your compliments and interest in this review.
With regard to reducing the technical specifications of the algorithms and devices, we have cut as much as possible.
Reviewer 2 Report
Comments and Suggestions for Authors
The authors through their review addressed how multi-parameteric algorithms that are available for cardiac devices whether the ICD and/or the CRT-D can predict a new heart failure event. As the use of AI tools in addition to clinical risk stratification tools especially in heart failure is an emerging field of work and so far many studies were promising. This article highlights this issue without ignoring the limitations of these tools. The way of exploring the existent algorithms was thorough and original. The discussion was informative.
The article summarized the existent algorithms and prior studies objectively, it discussed and showed the differences in different studies not only in terms of the used parameters but also differences in population involved which might affect the outcomes.
More elaboration of how data was extracted (chosen included studies) [e.g. any specific search terms] will be helpful.
Presentation of data was clear in terms of tables and figures.
Author Response
Comment: "More elaboration of how data was extracted (chosen included studies) [e.g. any specific search terms] will be helpful".
We conducted a review of remote monitoring systems for heart failure including clinical trials and real world studies performed. We used following search terms: "remote monitoring", "heart failure", "HeartLogic", "HeartInsight", "Triage-HF", "Cardiomems", "telemedicine".
A short description of how the data were searched has been added in the text.